# Phenotypes and Genotypes of Inherited Disorders of Biogenic Amine Neurotransmitter Metabolism

**DOI:** 10.3390/genes14020263

**Published:** 2023-01-19

**Authors:** Mario Mastrangelo, Manuela Tolve, Cristiana Artiola, Rossella Bove, Claudia Carducci, Carla Carducci, Antonio Angeloni, Francesco Pisani, Vincenzo Leuzzi

**Affiliations:** 1Child Neurology and Psychiatry Unit, Department of Human Neurosciences, Sapienza University of Rome, 00185 Rome, Italy; 2Azienda Ospedaliero Universitaria Policlinico Umberto I, 00161 Rome, Italy; 3Department of Experimental Medicine, Sapienza University of Rome, 00161 Rome, Italy

**Keywords:** neurotransmitter disorders, movement disorders, encephalopathy, dystonia, parkinsonism

## Abstract

Inherited disorders of biogenic amine metabolism are genetically determined conditions resulting in dysfunctions or lack of enzymes involved in the synthesis, degradation, or transport of dopamine, serotonin, adrenaline/noradrenaline, and their metabolites or defects of their cofactor or chaperone biosynthesis. They represent a group of treatable diseases presenting with complex patterns of movement disorders (dystonia, oculogyric crises, severe/hypokinetic syndrome, myoclonic jerks, and tremors) associated with a delay in the emergence of postural reactions, global development delay, and autonomic dysregulation. The earlier the disease manifests, the more severe and widespread the impaired motor functions. Diagnosis mainly depends on measuring neurotransmitter metabolites in cerebrospinal fluid that may address the genetic confirmation. Correlations between the severity of phenotypes and genotypes may vary remarkably among the different diseases. Traditional pharmacological strategies are not disease-modifying in most cases. Gene therapy has provided promising results in patients with DYT-DDC and in vitro models of DYT/PARK-SLC6A3. The rarity of these diseases, combined with limited knowledge of their clinical, biochemical, and molecular genetic features, frequently leads to misdiagnosis or significant diagnostic delays. This review provides updates on these aspects with a final outlook on future perspectives.

## 1. Background

Inherited defects of biogenic amine neurotransmitter metabolism are ultrarare genetically determined conditions resulting in dysfunctions/lack of enzymes involved in the synthesis, degradation, or transport of dopamine, serotonin, adrenaline/noradrenaline, and their metabolites or defects of their cofactor or chaperone biosynthesis (Table 1) [1]. All these conditions are inherited as autosomal recessive diseases, except for DYT/PARK-GCH1, including dominant and recessive forms and monoamine oxidase deficiency A and B, which are transmitted with an autosomal dominant inheritance [1]. 

Figure 1 summarizes the metabolic pathway and the sites of so-far identified proteins implied in clinically relevant alterations. Four enzymes (phenylalanine hydroxylase-PAH, tyrosine hydroxylase-TH, tryptophane hydroxylase-TPH, and aromatic aminoacidic decarboxylase-AADC) are directly implied in the synthesis of dopamine and serotonin. Five enzymes (guanosine triphosphate cyclohydrolase-GTPCH, 6-pyruvoyl tetrahydropterin synthase-PTPS, sepiapterin reductase-SR, dihydropteridine reductase-DHPR, and pterin-4a-carbinolamine dehydratase-PCD) are involved in the synthesis/regeneration of the PAH, TH, and TPH (tetrahydrobiopterin-BH4) cofactors. Three other disease-associated genes codify proteins acting as a chaperone of the three abovementioned hydroxylases (DNAJC12), an enzyme involved in biogenic amine catabolism (MAO), and a synaptic dopamine transporter (SLC6A3).

Several peculiarities justify the interest in these conditions: (a) as a group, they are among the most frequent genetic causes of movement disorders in children [1,2,3,4]; (b) many of them are treatable conditions, i.e., early therapy can bias the natural history of the disease, preventing severe neurological disabilities characterizing untreated patients [1,2,3,4]; (c) they have revealed the crucial function of serotonin and dopamine for the normal development of the central nervous system [1,2,3,4]. On clinical grounds, despite their relative rarity, each disease shares a unique pattern of recurrent clinical signs caused by dopamine and serotonin deficiency occurring in the immature brain [1,2,3,4]. From a biochemical viewpoint, they are considered a very early onset form of parkinsonism and parkinsonism-dystonia [1,2,3,4]. Thus, in this paper, all these diseases were named with the prefix “DYT/PARK” preceding the specific gene name, according to the recommendation of the Movement Disorder Society Task Force for the Nomenclature of Genetic Movement Disorders for genetic dopa-responsive dystonia/parkinsonisms [5]. 

This review provides updates on these genetic conditions, focusing on knowledge about genotype/phenotype correlations. 

## 2. Epidemiology

A reliable estimate of the frequency of this set of conditions is available for hyperphenylalaninemia-associated disorders, which are intercepted by neonatal screening programs for phenylketonuria (autosomal recessive DYT/PARK-GCH1, DYT/PARK-PTS, DYT/PARK-QDPR, and DNAJC12-related disorders), all including 2–3% of the overall number of patients with neonatal hyperphenylalaninemia (http://www.biopku.org/; last access 7 October 2022).

The first reported prevalence of autosomal dominant DYT/PARK-GCH1 was approximately 0.5–1.0 per million, with a remarkably higher penetrance of DYT/PARK-GCH1 variant carriers among females (87% vs. 38% in males) [6,7]. A more recent Serbian study estimated higher figures up to a prevalence of 2.96 per million [6,8]. A much lower prevalence of 1.4 per 100,000 patients under 18 was estimated in Estonia [9]. Autosomal recessive DYT/PARK-GCH1 probably affects less than 1 per 1,000,000 patients [3]. 

No adequate epidemiological data are available for other primary biogenic amine disorders. An indirect estimate of their frequency can be drawn on the basis of cases entered in the International Working Group on Neurotransmitter Related Disorders (iNTD patient registry (http://intd-online.org, last access 7 October 2022), which collected 350 patients with disorders of biogenic amine metabolism: 161 patients with BH4 deficiency, 56 patients with DYT/PARK-PTS, 37 patients with DYT/PARK-QDPR, 36 patients with autosomal dominant DYT/PARK-GCH1, 18 patients with autosomal recessive DYT/PARK-GCH1, 14 patients with DYT/PARK-SR, 131 patients with DYT-DDC, 44 patients with DYT/PARK-TH, 5 with DYT/PARK-SLC6A3, 5 with DNAJC12 deficiency, and 4 patients with MAO A deficiency [2].

The prevalence of DYT-DDC was also studied through other indirect evaluations [10,11]. The frequency analysis of 216 known pathogenic *DDC* variants (19 homozygous and 39 compound heterozygous) suggested a global prevalence of 1800 living patients with DYT-DDC [10]. An estimated prevalence of 1/42,000 live births per year was calculated for the same disease by analyzing biological samples from 19,684 American patients with neurological disorders of unknown origin [11].

## 3. Clinical Presentation

Table 1 summarizes the main clinical features of inherited biogenic amine neurotransmitter metabolism disorders. The onset of symptoms is in the first months or year of life for almost all these conditions, with two extremes represented by DYT/PARK-PTS, including the first possible manifestations also in the fetal life, and by autosomal dominant DYT/PARK-GCH1, with a prominent later presentation during school age or later [1,2,4]. An increased rate of prematurity or very low birth weight and congenital microcephaly was observed in DYT/PARK-PTS, DYT/PARK-TH, and DYT-DDC [2,12]. Exceptional late-onset presentations in adulthood have been reported for DYT/PARK-GCH1, DYT/PARK-SR, DYT/PARK-TH, DNAJC12, and DYT/PARK-SLC6A3 [2].

The pattern of neurological impairment, the age at the onset of symptoms, and their severity are usually correlated with the severity of biogenic amine depletion. The earlier the onset of the disease, the more severe and pervasive the neurological impairment is, resulting in a neurodevelopmental delay leading to intellectual disability, behavioral problems, and movement disorders [2,12,13,14]^.^ Speech and language impairment are typical clinical traits of late or untreated conditions and, similar to other higher cortical functions, they respond less than movement disorders to pharmacological treatments [2]. A more severe cognitive impairment has been reported in patients with DYT/PARK-SR, DYT/PARK-TH, and DYT-DDC [12,13].

The onset of symptoms after three is usually associated with the selective impairment of motor functions, as usually happens for autosomal dominant DYT/PARK-GCH1 [14]. An actual neurological regression was reported for DYT/PARK-SLC6A3 only, miming levodopa unresponsive degenerative parkinsonism [1]. 

The individual response to dopaminergic and serotoninergic treatment [1,14] may represent an additional source of clinical variability.

Movement disorders are a significant part of the clinical spectrum of inherited disorders of biogenic amine metabolism, both as an isolated disorder or in the context of infantile encephalopathy [14]. 

A remarkable source of complication for their management includes the lack of consensus for the classification of movement disorders in children under 3 years of life and the relevant differences between their clinical presentations and those typical of adult-onset parkinsonism [14].

Initial presentations may include rigidity of limbs and trunk hypotonia in patients with a global delay in developmental milestones [14]. Hypotonia is often associated with a dorsal trunk extension that might be considered a dystonic opisthotonus or an abnormal persistence of the fetal righting reflex [14]. 

In older children or adolescents, lead-pipe rigidity may often be combined with bradykinesia or focal arrhythmic jerks [14]. Akinesia and amimia may be difficult to assess in children because of higher interindividual variability of spontaneous motor activity and mimic expressions during the first months and years of life [14]. Classic akinetic-rigid syndrome may be preceded by a delay in antigravity motor development or pathological postural patterns [14]. Rest tremors are sporadic in early infancy. The whole semiology of rhythmic or pseudo-rhythmic oscillatory involuntary movements in infants cannot be categorized according to the tremor classification of tremors in adults [14].

## 4. Defects of Synthesis of Biogenic Amines

This group of disorders includes diseases with hyperphenylalaninemia (detectable with newborn screening) and diseases without hyperphenylalaninemia (Table 1).

### 4.1. Diseases with Hyperphenylalaninemia

#### 4.1.1. Autosomal Recessive DYT/PARK-GCH1

This autosomal recessive disease is due to pathogenic homozygous or compound heterozygous variants of the *GCH1 (NM_000161.3)* gene. *GCH1* (14q22.2, OMIM*600225) includes six exons and encodes for the GTPCH1 enzyme that catalyzes the first step in BH4 biosynthesis (the production of D-erythro-7,8-dihydroneopterin triphosphate from GTP) [3] (Figure 1). 

The ClinVar database reports 360 variants (https://www.ncbi.nlm.nih.gov/clinvar/; last access 7 October 2022). Five of them are structural pathogenic/likely pathogenic variants involving other genes contiguous to *GCH1*, whereas another reported that the pathogenic complete deletion of the *GCH1* gene leads to haploinsufficiency [15]. The most frequently reported pathogenic/likely pathogenetic variants were missense mutations (*n* = 28), followed by frameshift (*n* = 18), splice site (*n* = 16), and nonsense ones (*n* = 11). The number of pathogenic variants reported in ClinVar drops to 56 after filtering, excluding those deposited without assertion criteria by submitters (Appendix A). Missense, frameshift, and splice site variants were equally distributed. In contrast, nonsense ones were less frequently reported (Appendix A).

The BioPKU database reports 329 *GCH1* gene variants, but only a minor proportion of them (*n* = 42) are marked as pathogenic/likely pathogenic. Several variants associated with a suggestive clinical and biochemical phenotype in the literature or with a demonstrated disease-causing role after specific functional studies were not reviewed and reassessed in the two databases above. The recent review by Himmelreich et al. reported seven variants associated with clinical manifestations of *GCH1* [6]. The interpretation of their effects is only sometimes homogeneous between ClinVar and BioPKU (Table 2).

The c.350T > G (p.Leu117Arg) variant, which probably interferes with the access of GTP to its binding site, caused severe early onset motor dysfunctions in a recently developed homozygous mouse model mutant [16]. This variant had been previously associated with an intermediate phenotype in a patient presenting with a heterozygous transmission [17]. Very few cases of compound heterozygous autosomal recessive DYT/PARK-GCH1 without hyperphenylalaninemia were also reported [17].

Dystonia and tone abnormalities were reported as the initial signs in a proportion ranging between 25 and 40% of patients [2,18,19,20]. Early presentation includes rigidity and tremulous or intermittent jerky movement of limbs during the neonatal period, evolving in a few months towards generalized dystonia with prolonged dystonic spasms, limb spasticity, and an absence of postural reaction development, oculogyric crisis, hypomimia, and poor sucking followed by severe feeding difficulties in the subsequent stages [18,19,20]. Diurnal fluctuation of dystonia was sporadically observed [21]. Early developmental delays or stagnation followed by increasing rigidity of limbs with trunk hypotonia and a lack of postural reaction development have been rarely observed [22,23]. A later presentation of progressive dystonia with diurnal fluctuations has been occasionally reported [24].

#### 4.1.2. DYT/PARK-PTS

DYT/PARK-PTS is an autosomal recessive disease due to pathogenic variants of the gene that is located on chromosome 11q22-3-q23.3 [7]. The encoded enzyme catalyzes the removal of triphosphate from the substrate 7,8-dihydroneopterin triphosphate (Figure 1) [7]. 

Most reported pathogenic variants of *PTS* were missense, while splice sites, nonsense, and frameshift ones were less frequently described (Appendix A).

The ClinVar database (last access 7 October 22) reports 212 variants involving *PTS,* with a pathogenic duplication on several genes located in the region q22.1-25 and a deep 55bp intronic deletion in intron 2 (c.163 + 696_163 + 750del), which result in a pseudoexon [25].

The BioPKU database reports 198 variants, including a multiexonic deletion whose breakpoint has not been characterized [21].

A probable founder effect was suggested for variants c.155A > G (p.Asn52Ser), c.259C > T (p.Pro87Ser), c.272A > G (p.Lys91Arg), c.286G > A (p.Asp96Asn) (Appendix A), c.84-291A > G, c.58T > C (p.Phe20Leu), and c.243G > A (p.Glu81 = ) in patients coming from different Asian regions, variant c.238A > G (p.Met80Val) (Appendix A) in the Arab population, and variant c.317C > T (p.Thr106Met) in Russian patients (Appendix A, Table 2) [7].

A severe (85% of patients) and a mild phenotype (15% of patients) of DYT/PARK-PTS were defined according to the clinical presentation and biochemical alterations [26]. Variants c.46C > T (p.Arg16Cys), c.78G > T (p.Leu26Phe) (Appendix A), c.338A > G (p.Tyr113Cys), and c.370G > T (p.Val124Leu) are associated with the mild phenotype [7]. Variants c.139A > G (p.Asn47Asp), c.347A > G (p.Asp116Gly), and c.412A > C (p.Asn138His) are associated with isolated and transient hyperphenylalaninemia [7]. Variants c.120T > G (p.Phe40Leu), c.216T > A (p.Asn72Leu), and c.430G > C(p.Gly144Arg) are associated with isolated hyperphenylalaninemia but also with more severe phenotypes [7]. 

Recombinant expression and Western blot analyses in patients’ fibroblasts revealed the complete inactivation or the structural destabilization of the PTPS enzyme associated with variants c.200C > T/p.Thr67Met (Appendix A), c.385A > G/p.Lys129Glu, and c.407A > T/p.Asp136Val (Appendix A) [24]. Inhibited access to sites for the regulatory phosphatase, resulting in an incompletely active enzyme, was correlated with the variant c.46C > T/p.Arg16Cys [7].

Movement disorders were reported in about 25% of children with the severe phenotype [26]. The most common early motor disorders include axial hypotonia or generalized hypertonia, hyperreflexia, tremors of the upper limbs, spasticity of limbs, and poor sucking during the first weeks of life [26,27,28]. These symptoms might result from premature birth and low birth weight occurring in many patients with DYT/PARK-PTS [26,27,28]. 

Different patterns of clinical progression may be assessed after the neonatal-infantile period: (1) slowing of weight gain and developmental delays, diffuse dystonia with opisthotonus associated with lead-pipe rigidity or hypotonus, myoclonic jerks, and ataxia [26,29,30,31]; (2) rigid akinetic syndrome with hyperreflexia and oculogyric crisis [29,32]; and (3) ataxic-dystonic cerebral palsy [21]. These different conditions may result in a hypotonic-akinetic or rigid-akinetic syndrome with diffuse myoclonic jerks and oculogyric crisis [32]. Paroxysmal dyskinesias during the first year of life have also been reported [29]. A BH4 responsive restlessness, pallor, ptosis, increasing hypotonia, and ataxia evolving in a semi-comatose state were observed in a young girl [29]. Her brother suffered from short dystonic episodes of opisthotonus with limb rigidity and clonus of the feet at the age of 6 months [29]. A transient improvement of dyskinesia was obtained with the administration of L-dopa/carbidopa and 5-hydroxytryptophan [29].

Psychiatric disorders were recently reported in 39% of cases in an Italian cohort, including 28 patients [33]. 

#### 4.1.3. DYT/PARK-QDPR

DYT/PARK-QDPR is an autosomal recessive disease due to pathogenic variants of the cited gene on chromosome 4p15.32 (NM_000320.3, OMIM*612676) [3,7]. The homodimer-encoded enzyme (DHPR) catalyzes the NADH-mediated reduction of quinonoid dihydrobiopterin (Figure 1) and is an essential component of pterin-dependent aromatic amino acid hydroxylating systems [34]. A total of 141 pathogenic variants distributed all over the seven exons of the gene were reported in the literature [7]. 

ClinVar and BioPKU include, respectively, 257 and 141 variants involving *QDPR*. No clear predominance of specific types of point mutations was highlighted, whereas 13 large deletions or multigenic duplications and an inversion of the entire gene were also deposited [35].

Most reported variants are associated with inactive enzymes, with a few exceptions correlated with milder phenotypes. The variants c.451G > A(p.Gly151Ser) and c.635T > G(p.Phe212Cys) are associated with isolated abnormalities of serotonin metabolism, whereas c.199-1G > T is associated with an isolated mild hyperphenylalaninemia [7]. 

DHPR loss or reduced function results in severe brain damage that might be caused by two main mechanisms: (1) the secondary accumulation of 7,8 dihydrobiopterin (BH2) and the subsequent inhibition of aromatic amino acid hydroxylases causing a low concentration of dopamine and serotonin in the brain; and (2) the reduced availability of 5-methyl-tetrahydrofolate because of a secondary defect of brain dihydrofolate reductase diverted to the synthesis of BH4 from BH2 to BH4 [7,36]. 

The clinical presentation of DYT/PARK-QDPR is characterized by developmental delays, intellectual disability, and epilepsy, while movement disorders occur less frequently (less than 20% of infants and 30% of older children) [37,38]. In untreated or late-treated patients, the early pattern of neuromotor impairment includes oculogyric crisis, tremors, hypotonia, hyposthenia, and developmental delays [37]. Muscle hypotonia is observed in approximately 10% of newborns and 50% of infants and children [37]. A few patients with this disease experienced stroke-like episodes despite early treatment and adequate metabolic control [39]. In the few documented cases, strokes were associated with a deficient level of 5-MHTF in CSF, suggesting a possible disease-causing role of folate depletion [39]. 

#### 4.1.4. Pterin-4a-Carbinolamine Dehydratase Deficiency 

Pterin-4a-carbinolamine dehydratase deficiency is caused by homozygous or compound heterozygous variants of the *PCBD1* (NM_000281.4, OMIM*126090) gene on chromosome 10q22.1, which encodes an enzyme involved in the dehydration of pterin-4a-carbinolamine to quinoid dihydropteridine. (Figure 1). No severe neurological symptoms and transient and benign hyperphenylalaninemia are the reported clinical hallmarks [40]. 

ClinVar reports 68 items, including considerable deletion/duplication with no associated phenotypes, whereas BioPKU includes only 32 variants. The few variants with provided assertion criteria result in a conflict of interpretation about the disease-causing role or are deposited as VOUS (Appendix A).

### 4.2. Diseases without Hyperphenylalaninemia

#### 4.2.1. Autosomal Dominant DYT/PARK-GCH1

*GCH1* variants are distributed all over the parts of the genes, many of which are in exon 1 (Appendix A) and are associated with broad phenotypic variability [7]. Some studies excluded significant genotype–phenotype correlations. In contrast, others reported an association with multifocal dystonia occurring in Taiwanese adult patients carrying different large heterozygous *GCH1* gene deletions with high penetrance [6]. A probable founder effect has recently been suggested for variant c.265C > T/(p. Gln89*) in Andalusia [8].

Clinical manifestations are associated with heterozygous variants resulting in residual GTPCH activity lower than 20% or destabilized enzyme structures [7]. Higher expressions of heat shock proteins/molecular chaperones (HSP70 and HSP90) acting on protein misfolding prevented the reduction in GTPCH enzyme levels in an old study [41]. A completely lacking enzyme activity is associated with variants c.262C > T(p.Arg88Trp) and c.551G > A(p.Arg184His) (Appendix A) [7]. 

A dominant negative disease-causing effect was suggested for some pathogenic variants in a heterozygous state in which a chimeric destabilized protein, including the wild type and variants, subunit was synthesized: c.633G > A, p.Met211Ile (Appendix A), c.626 + 1G > A (Appendix A), and c.557C > A (p.Thr186Lys) [6]. Another proposed mechanism included a low enzyme expression level associated with the co-expression wild type/destabilized pathogenic variant [7]. A stabilizing effect was demonstrated for specific heat shock proteins (HSP70 and HSP90) [7].

The mean age at the onset of motor symptoms is about 6 years, whereas movement disorders with onset during infancy and early childhood are extremely infrequent (up to 1 of 20 patients) [42,43]. Diurnal fluctuation of symptoms is typical in these cases, especially for dystonia, with a remarkable worsening during the day and a substantial improvement after sleep [39]. Other frequent symptoms in the later stages may include parkinsonism, sleep disturbances, and neuropsychiatric symptoms, such as anxiety and depression, while intellectual disabilities were reported in a small proportion of patients [6]. 

Early clinical onset is presumably underdiagnosed (diagnostic delay is longer than 10 years in several cases), and about 20–30% of patients are symptomatic within the age of 3 years [44,45,46,47]. The few reported early clinical presentations included:-Neuromotor and cognitive regression with loss of head control, feeding disturbances, and diffuse athetoid movements [48];-Delay in early motor milestones with pyramidal and extrapyramidal signs mimicking a cerebral palsy [49,50,51];-Diurnal fluctuation of balance and tremors of limbs [49];-Hypokinesia and drooling at the age of 9 months [50];-Intermittent and fluctuating tiptoe walking evolving into gait difficulties and paroxysmal opisthotonus with akinesia lasting 2 to 3 h [51];-Lower limb focal dystonia occurring between 2 and 3 years [52,53].

In our case, mild developmental delays during the first 2 years led to myoclonic jerks by the age of 3 years and myoclonic dystonia during adolescence in a boy carrying the variant c.671A > G (p.Lys224Arg) who had myoclonus-dystonia and resting or postural tremors of limbs in his paternal branch [54].

#### 4.2.2. DYT/PARK-SR 

DYT/PARK-SR is an autosomal recessive disease caused by pathogenic variants that may be distributed along the 3 exons of the gene (NM_003124.5) on chromosome 2p13.2. SR regulates the reduction of the pterin intermediate 6-pyruvoyl-tetrahydropterin included in de novo BH4 biosynthesis (Figure 1).

Most of the 155 variants deposited on ClinVar are classified as VOUS, while very few are marked as pathogenic with provided assertion criteria and compatible associated clinical phenotypes (Appendix A).

The compound heterozygous state involving variants c.512G > A(p.Cys171Tyr) and c.304 + 1_ + 12del (marked in Appendix A as c.304 + 2_304 + 13del) destroying the 5′ splice donor site in intron 1, with subsequent intron retention, was detected in a patient with L-dopa responsive dystonia [55]. 

A possible founder effect was suggested for c. 68G >A(p.Gly23Asp) in the Maltese population [56]. The heterozygous variant c.207C > G(p.Asp69Glu) is associated with an autosomal dominant dopa-responsive dystonia and incomplete penetrance in five members of a family in which increased levels of urinary sepiapterin were also detected [57].

Variant c.-13G > A, which is in the 5’UTR, was described in a patient with autosomal dominant responsive dopa dystonia. Functional studies on patients’ fibroblasts showed a decrease in SR enzyme activity and the amount of synthesized protein [58]. 

The onset of symptoms ranged between 7 months and 6 years in a 43-patient multicentric retrospective study [59]. Early clinical presentations include motor delay, axial hypotonia, oculogyric crisis, developmental delays, diurnal fluctuations, dystonia associated with limb hypotonia or hypertonia, swallowing impairment, chorea, and ataxia [59]. Axial hypotonia and developmental delays were reported in 92% of patients before 4 years of age, while 40% of patients presenting with dystonia experienced the symptoms since early infancy [59,60]. Diurnal fluctuations of motor symptoms appeared during infancy in 60% of patients [59]. A mean diagnostic delay of 9.1 years in the diagnosis, which was still compatible with a relevant improvement of movement disorders in almost all the patients under L-dopa/carbidopa treatment, was detected [59]. A relevant history of motor derangement was defined in two early-diagnosed cases with hypokinetic-rigid syndrome with impairment of the development of the postural reaction, episodes of spasmodic dystonia of the trunk with oculogyric crises and oral tongue dyskinesia, and tremors of the limbs and head at rest that could be inhibited by skin contact and/or spontaneous movement [60,61]. This last presentation, which is uncommon in infancy, resembles rest tremors within the akinetic-rigid syndrome in adults [61,62].

#### 4.2.3. DYT/PARK-TH 

DYT/PARK-TH is an autosomal recessive disease caused by pathogenic variants of the gene (NM_000360.4, OMIM*191290) on chromosome 11p15.5 [62]. The *TH* gene spans about 8 Kb and includes 14 exons [63]. The encoded homotetrameric enzyme converts L-tyrosine to L-3,4-dihydroxyphenylalanine (L-DOPA), the essential and rate-limiting step to the formation of dopamine and other catecholamines [63]. *TH* is predominantly expressed in dopaminergic neurons of areas regulating motor functions (nigrostriatal networks) but also motivation, addiction, and reward (connections between the ventral tegmental area, nucleus accumbens, and pre-frontal cortex) [63]. 

A pathogenic role was reported for 66 variants, including 14 nonsense mutations and 19 mutations in the promoter [63,64]. 

ClinVar reports 64 variants of the *TH* gene with a demonstrated pathogenic role (23 with assertion criteria including 10 nonsense, 9 frameshift, 3 missense, and 1 regulatory variant) and 237 VOUS. No clear genotype–phenotype correlations are demonstrated [63,64]. Seven variants were reported with a higher frequency (Table 2) [64].

Two partially overlapping phenotypes of tyrosine hydroxylase deficiency were suggested in a 2010 retrospective study: in the former (type A), motor functions are more selectively involved, while the latter (type B) identifies an “encephalopathic” form, including a minority of children with a more generalized impairment of neurological functions and development [65]. The recent iNTD registry analysis concluded that the type A/B classification is not clinically justified because of many overlapping presentations and proposed abandoning it [2,65]. 

Patients with mild phenotypes usually present with developmental delays, generalized hypokinetic-rigid syndrome, and fluctuating dystonia during the first weeks or months of life [65]. Dystonia generalizes with a centripetal pattern involving the face or pharyngeal muscles in late infancy presentations [65]. In these cases, psychomotor development and mental functioning are preserved or may be slightly delayed [65]. 

One of our patients presented with normal psychomotor development until the second year of life when toe-walking gait, frequent falls, and delayed language suggested the diagnosis of spastic paraplegia [65]. Later, at the age of 5, oculogyric fits were misdiagnosed as epileptic seizures [65]. Finally, at the age of 11, progressive posture control instability and choreoathetosis suggested the final diagnosis of DYT/PARK-TH [65]. Patients with mild phenotypes usually respond to pharmacologic treatments with L-dopa or dopamine agonists [65]. Choreiform movements and paroxysmal dyskinesia are side effects of the pharmacologic treatment because of the hyper-elicited postsynaptic response to dopamine and D2 and D1 receptor supersensitization [65]. These effects can be transiently observed for a few weeks after the beginning of the treatment and in the following stages according to the clinical and biochemical severity of the phenotype, the administered dosage, and the rapidity of the dose increase [65,66,67,68].

Patients with more severe phenotypes display an early derangement of neurological development, a severe hypotonic-hypokinetic syndrome with dystonic movements and posture, oculogyric crises, ptosis, focal or generalized dystonic features, and jerky movements such as tremors and myoclonus [65]. The onset of the disease during the first weeks of life may be triggered by an intercurrent disease mimicking encephalitis [65]. Worsening oculogyric crisis and tremors lasting several hours and occurring every 2 to 3 days may precede the occurrence of neurological deterioration and continuous movement disorders [65].

Sometimes, severe hypokinesia and hypotonia may remind the patterns of muscle diseases or spinal muscle atrophy^65^. Fluctuations in neurological symptoms include a periodical worsening of dystonic crises or lethargy-irritability crises [65]. Patients with more severe phenotypes have a worse prognosis than patients with mild phenotypes due to L-dopa tyrosine hydroxylase hypersensitivity [65]. In index patient phenotyping, for the previously called “type B pattern”, an extremely gradual dosage rise of L-dopa/carbidopa induced a significant motor improvement with the achievement of autonomous gait at the age of 10 years, while peak dose dyskinesia occurred after each increase in the dosage and during periods of the emergence of new gross motor skills [69]. 

Several reported patients presented with complex phenotypes [2,65,70]. At the age of 3 months, one of our patients presented with a stagnation of psychomotor development and spontaneous and evoked fits (stare and upper-limb jerks) [70]. At the age of 6 months, he was an alert and reactive child with poor somatic growth, severe developmental delays, hypokinesia, trunk hypotonia, limb rigidity, and action tremors [70]. By 13 months, the impairment of mental development was severe, and the patient suffered from paroxysmal dystonic postures and movements of limbs associated with severe diffuse hypokinesia [70]. Treatment began at 18 months and resulted in a rapid remission of movement disorders and improved motor development [70]. A recently reported 6-month-old child presented with floppiness and poor head control which evolved into severe myoclonic jerks of the limbs, dystonia without diurnal fluctuations, and protracted painful diffuse dystonic spasms involving the limbs and trunk [71]. Dystonic spasms were triggered by fever, infections, or tiredness and occurred several times daily in association with oculogyric crises and double incontinence [71].

#### 4.2.4. DYT-DDC 

DYT-DDC is an autosomal recessive inherited disease due to pathogenic gene variants (NM_001082971.2, OMIM*107930) on chromosome 7p12.2-p12.1 [72]. AADC is a homodimeric pyridoxal 5′ phosphate (PLP)-dependent enzyme involved in two main metabolic pathways: the decarboxylation of L-DOPA to synthesize dopamine and the decarboxylation of 5-OH tryptophan to form serotonin [72]. 

The BioPKU database includes 422 variants: 7% of them are identified as pathogenic, 32% as likely pathogenic, 58% as VOUS, 1% as likely benign, and 1% as benign [72]. A total of 108 genotypes were reported in patients with confirmed molecular diagnosis [72]. Genotype–phenotype correlations were confirmed in most cases, with a prominence of missense variants over frameshift and nonsense ones (Appendix A) [72,73]. A founder effect was suggested for variants c.714 + 4A > T (IVS6 + 4A > T), c.1297dup (p.Ile433Asnfs*60), and c.1234C > T (p.Arg412Trp) in the Taiwanese population (Appendix A) [72,74]. 

Variants c.304G > A (p.Gly102Ser), c.1040G > A(p. Arg347Gln) (Appendix A), and c.478C > T(p.Arg160Trp) were detected in patients responsive to L-dopa treatment, while variant c.749C > T (p.Ser250Phe) was predicted to be associated with responsiveness to pyridoxine supplementation by in vitro studies (Appendix A) [74,75].

The variant c.304G > A(p.Gly102Ser) was found with an allele frequency of 8% of patients in a 78-patient cohort and is associated with an AADC enzyme activity of 16% when compared to the wild-type protein [76]. In the same cohort, the allele frequency was 45% for the c.714 + 4A > T variant and 10% for variant c.749C > T [76].

The clinical presentation of DYT-DDC includes combined signs of dopaminergic (dystonia, oculogyric crises, and hypokinesia), serotoninergic (sleep disorders, memory, and learning disabilities), and noradrenergic (orthostatic hypotension, temperature dysregulation, and ptosis) dysfunctions [74]. Symptoms fluctuate during the day and improve after sleep [73]. Developmental delays are usually observed in most patients [74]. About 80% of published patients present with severe motor impairments, whereas mild phenotypes include preserved autonomous gait [74]. 

Early movement disorders, including delay in postural reactions with severe hypotonia and various dystonic presentations with a prominence of oculogyric crises, occur in early infancy in 95–97% of cases aged 2–12 years [74].

The index patients, i.e., two male monozygotic twins born to related parents at 2 months, presented with severe hypotonia, crying-induced dystonic postures of the arms and the legs, oculogyric crises, and occasional choreoathetosis movements of the extremities [77]. The following observations have expanded the phenotype of movement disorders in DYT-DDC to brady/akinetic and asthenic patterns that, during the first year of life, may resemble myasthenia condition [77,78,79,80]. Hypotonia can be associated with dystonic reactions after external stimuli [81]. In some patients, dystonic postures can be followed by a startle myoclonus [81]. Hyperkinetic movements, including choreoathetosis and tremors, are less frequent in patients under 3 years [74,76]. 

Clinical observations in two of our female patients with mild symptoms evidenced a spontaneous improvement of motor symptoms occurring during the transition from the early stages of life into adolescence and young adulthood [80]. One of the two patients experienced two pregnancies without significant problems [82,83]. 

A recent retrospective study in a 9-patient Italian cohort evidenced a susceptibility to psychopathological and psychiatric disturbances, including generalized anxiety disorder, depressive disorder, obsessive–compulsive disorder, oppositional defiant disorder, and attention deficit hyperactivity disorder [84].

## 5. Defects of Biogenic Amine Catabolism

### 5.1. Monoamine Oxidase (MAO-A/MAO-B) Deficiency

The two isoenzymes monoamine oxidases A and B, located in secretory neurotransmitter vesicles, are key enzymes responsible for the oxidative deamination of biogenic amines [85]. Genes encoding for the two isoenzymes (MAO-A:NM_000240.4, MAO-B:NM_000898.5) are sited on chromosome Xp11.23 [84]. Both genes are expressed in many tissues, including the brain, liver, and kidneys [85].

MAOA deficiency presents with intellectual disabilities, autonomic dysregulation, and behavioral disturbances [85]. MAO-B deficiency was reported in subjects without intellectual and behavioral impairments [85]. Combined MAO A and MAO B deficiency phenotypes include intellectual disabilities, recurrent stereotypes, sudden loss of muscle tone, and epileptic seizures [86]. 

Most pathogenic variants deposited in ClinVar include large duplications involving many genes and extending more than 1 kb, with only a few mutations smaller than 50 bp (Appendix A). The de novo variant c.1336G > A (p.Glu446Lys) in a patient with MAOA deficiency was associated with increased serum serotonin and urinary serotonin, norepinephrine, epinephrine, dopamine, metanephrine, and normetanephrine [87]. Negligible abnormalities of MAO activity were detected in three hemizygous patients carrying the variant c.886C > T(p.Gln296*) [87].

### 5.2. Dopamine-β-Hydroxylase Deficiency

Dopamine-β-hydroxylase deficiency (DBH-D), encoded by the *DBH* gene (NM_000787.4, OMIM*609312) on chromosome 9q34.2, is a membrane-bound enzyme that converts dopamine to norepinephrine in presynaptic vesicles. To date, 25 patients from 20 families are described in the literature, mainly in the USA and the Netherlands [88]. 

The molecular genetic diagnosis was achieved in 19 patients from 16 families, and 10 different pathogenic variants were identified [88]. A compound heterozygous state was reported in 10 patients [88]. The intronic variant c.339 + 2T > C (Appendix A), resulting in aberrant splicing, was detected in 19 patients (7 of them had a homozygosis) [88]. Many variants deposited in ClinVar involve other contiguous genes with subsequent structural effects. 

Severe orthostatic hypotension, eyelid ptosis, sporadic dysmorphic features, rare reproductive dysfunctions, and normal cognitive development represented the most common clinical hallmarks [88]. 

## 6. Defects of Transport of Biogenic Amines

### 6.1. DYT/PARK-SLC6A3 

The dopamine transporter (DAT), encoded by the *SLC6A3* gene (NM_001044.5, OMIM*126455) on chromosome 5p15.33, modulates the active reuptake of dopamine from the synapse and is a principal regulator of dopaminergic neurotransmission [89,90,91]. Loss of DAT function results in the accumulation of dopamine in the synapse followed by a reduced dopamine reuptake and recycling and by desensitization of dopamine receptors [89,90,91]. 

DYT/PARK-SLC6A3 is an autosomal recessive disease that was diagnosed in 14 published patients [91,92,93]. No variants were shared among unrelated patients suggesting that no mutational hotspots exist [91]. 

ClinVar reports only six pathogenic nonstructural (>50 bp) variants of the *SLC6A3* gene, and only three of them were deposited with assertion criteria (Appendix A). Most of the deposited variants are classified as VOUS. The homozygous variant c.1561C > T(p.Arg521Trp) is the only deposited mutation associated with a suggestive clinical phenotype (a patient presenting with oromandibular dystonia, bilateral striatal toes, recurrent episodic dystonic crises, and a good response to levodopa) [90]. 

Other recently reported likely pathogenic variants, which were not deposited in ClinVar, include c.934A > T (p.Ile312Phe), causing a structural perturbation with an indirect impact on catalytic activity, c.1261G > A (p.Asp421Asn), compromising the sodium binding site, and c.1639dupC (p.His547Profs*56), which influences the structure of the transporter.

DYT/PARK-SLC6A3 manifests during the first months of life with two of the following alternative patterns of neuromotor derangement: (1) a severe akinetic-rigid syndrome and (2) a severe dyskinetic syndrome [90,91,92].

Among all the different monoamine neurotransmitters related to movement disorders, this is probably the only disorder manifesting with such extreme and precocious hypokinesia associated with remarkable impairment of gross motor and intentional skill development [91,92,93]. Four out of seven patients in the index cohort presented akinetic–rigid syndrome within 12 months from the onset of the disease [90,91]. In a few patients, akinesia was associated with postural persistence such as catatonia, which was fragmented by continuous positive and negative multifocal myoclonic jerks of the limbs [90,91]. 

Hypomimia is particularly remarkable in DYT/PARK-SLC6A3, sometimes contrasting with an incongruous and persistent motionless smile and with oculogyric crises or other ocular dyskinesia, such as ocular fluttering and opsoclonus coupled with an increase in saccade latency [90]. In the first reported cohort, hypomimia was noticed at a mean age of 7 months (range: 2–18 months) [90,91]. In some patients, akinetic-rigid syndrome is associated with focal or segmental dystonic posture [90,91]. A severe dyskinetic syndrome occurred at the beginning or later stages of the disease in some patients [90]. It is characterized by continuous, asynchronous, and diffuse myoclonic jerks in some patients and compulsive, stereotyped, and, in some way, patterned (boxing-like) ballistic movements of all limbs in others [90]. In addition, this dyskinetic presentation is associated with hypomimia [90]. The clinical progression of the disease is toward a rigid, akinetic, and flexed posture, also in DAT-deficient patients presenting with early-stage hyperkinetic movement disorders [91]. The reported developed postural reactions are not compatible with sitting, standing up, or walking [90,91,94]. Later-onset phenotypes include a few cases of adult-onset dystonia-parkinsonism, ADHD, and seizures [95,96].

### 6.2. DYT/PARK-SLC18A2 

DYT/PARK-SLC18A2 is an autosomal recessive disease affecting a protein involved in the transport of biogenic amines inside the presynaptic vesicles [97]. The encoding gene is *SLC18A2* (NM_003054.4, OMIM*193001) on chromosome 10q25.3. 

The index family included eight affected members [97]. All the cases carried the variant c.1160C > T (p. Pro387Leu) [96]. In vitro functional expression studies in COS-7 cells showed that the variant resulted in a severe, albeit incomplete, loss of transport function [97]. The index case presented at 4 months with hypotonia, no head control, oculogyric crises and crying, akinesia and hypomimia, paroxysmal dystonia without diurnal fluctuations, and global developmental delays [97]. Later in childhood, the disorder evolved into a clinical pattern resembling Parkinson’s disease with dyskinesia [97]. This condition responded to dopamine-mimetic drugs but not to biogenic amine precursors [9].

More recent studies extended the total number of patients to 58 affected individuals (16 of them died before the age of 13) carrying 19 homozygous variants and presenting with global developmental delays, hypotonia, dystonia, oculogyric crises, and autonomic symptoms [98,99,100,101,102]. A weak relationship between some variants and the phenotypic severity was suggested [102]. Patients carrying c. 710C > A (p.Pro237His) died in nine cases out of seventeen, while a milder phenotype and prolonged survival was observed in subjects carrying c.127A > T (p.Ile43Phe) and c.1160C > T (p. Pro387Leu). [102].

## 7. Disorders of Chaperone Molecules

### DNAJC12 Deficiency (OMIM # 617384)

The recently reported DNAJC12 deficiency is caused by autosomal recessively transmitted pathogenic variants of the *DNAJC12* gene (NM_021800.3, OMIM*606060) on chromosome 10q21.3. DNAJC12 is a heat shock cochaperone protein interacting with PAH, TH, and TPH [103]. An impaired folding of these three enzymes probably results in hyperphenylalaninemia (detectable via neonatal newborn screening) and signs of central dopamine and serotonin deficiency without the typical pattern of BH4 deficiency [103]. The coexistence of variants in *DNAJC12* and *PAH* in some Spanish patients suggested the possible role of DNAJC12 as a modifier of PAH deficiency [104]. In this context, different DNAJC12 variants may stabilize/degrade PAH, and these alterations support the necessity to re-evaluate cases of misdiagnosed phenylketonuria in which genotype–phenotype correlations are inconsistent [104]. 

Forty-one patients were reported in the literature, aged between 2 and 40 years [105,106]. 

The clinical phenotype is highly variable, including different degrees of hyperphenylalaninemia, juvenile parkinsonism, dystonia, autism, intellectual disabilities, attention deficit hyperactivity disorder, psychiatric symptoms, and no symptoms in 18 patients [107,108]. 

Twelve pathogenic nonstructural variants with direct and demonstrated effects on phenylalaninaemia were deposited in ClinVar. Most of them were inserted without assertion criteria (Appendix A). Other variants that are not consistently correlated with hyperphenylalaninemia include c.306C > G(p.His102Gln) and c.182delA(p.Lys61Argfs*6), c.85delC(p.Gln29Lysfs*38), c.596G > T( p.Ter199Leu) and c.214C > T(p.Arg72*), and NC_000010.11:g.67799813_67806755del(c.298-968_503-2603del) (Appendix A) [108].

## 8. Diagnostic Work-Up 

Figure 2 indicates a suggested flowchart for diagnosing primary defects of biogenic amine metabolism. Table 2 summarizes the interpretation of the variants with more significant reported clinical implications according to the databases ClinVar and BioPKU. 

Several biochemical markers (blood phenylalanine, urinary pterin, and cerebrospinal fluid measurements of biogenic amine metabolites and pterins) are essential for a correct address to genetic confirmation (Figure 2 and Table 1). The measurement of homovanillic acid (HVA), 3-O-methyldopa (3OMD), 3-methoxy4hydroxyphenylglycol (MHPG), 5-hydroxy indoleacetic acid (5HIAA), neopterin, tetrahydrobiopterin (BH4), dihydrobiopterin (BH2), 5-methyltetrahydrofolate, and pyridoxal 5’phosphate in cerebrospinal fluid is performed with high-performance liquid chromatography in a few specialist centers worldwide after having followed a specific procedure for sample collection [1]. Neonatal dried blood spot (DBS) screening tests and plasma amino acid analysis may reveal hyperphenylalaninemia associated with pterin defects [1,3,4]. The detection of increased levels of 3-orto-methyldopa (3OMD) and 5HTP offer new reliable biomarkers for the newborn screening of DYT-DDC [108,109,110,111]. Hyperprolactinemia may be used as an indirect, although not specific, marker of central dopaminergic deficiency because of the physiological inhibitory role of prolactin towards dopamine [1,3,4]. The measurement of amino acid levels after an oral loading test with BH4 or phenylalanine may provide relevant data for the differential diagnosis between BH4 deficiencies, phenylketonuria, and other causes of dystonia [4]. 

The measurement of urine levels of neopterin, biopterin, and sepiapterin may detect pterin defects, while high urine levels of vanillylactic acid may reveal DYT-DDC [3].

In some cases, biochemical markers may correlate with the clinical severity with possible prognostic implications. CSF HVA and HVA/5HIAA ratios significantly correlated with more severe phenotypes in a cohort of 36 patients with DYT/PARK-TH [65]. Patients with mild phenotypes of DYT-DDC carrying a compound heterozygous state for c.105delC/c.710 T > C presented with normal CSF HVA levels and a marginal reduction in CSF-5HIIA [80]. Severe forms of DYT/PARK-PTS are associated with a CSF profile including low levels of HVA and 5-HIAA and an altered neopterin/biopterin ratio, while mild forms presented with normal HVA and 5-HIAA and an altered neopterin/biopterin ratio [33].

A recent proteomic study on the CSF of 90 patients revealed novel possible severity biomarkers, including decreased levels of apolipoprotein D for DYT-DDC, a decrease in apolipoprotein H, an increase in oligodendrocyte myelin glycoprotein for tyrosine hydroxylase deficiency, and a decrease in collagen6A3 for BH4 deficiency [112].

Brain neuroimaging is often expected, even if a recent systematic re-evaluation of 87 MRIs of 70 patients evidenced different abnormalities: diffuse cortical/subcortical atrophy in 24 patients, T2-hyperintensity of central tegmental tracts in 9 patients, changes in arterial watershed zones involving parieto-occipital and cerebellar areas in 8 patients (DYT/PARK-QDPR), myelination delay in 4 patients (DYT-DDC and DYT/PARK-TH), signs of profound hypoxic-ischemic injuries, and a combination of deep gray matter and watershed injuries in 2 patients (DYT-DDC) [113]. Volumetric studies on 15 patients evidenced an association of total volume deficits of cortical and subcortical structures with clinical severity, independently of the diagnosed diseases [114]. 

The loss of DAT activity in the basal ganglia, as measured by single-photon emission tomography of DAT, may be a useful diagnostic marker in patients with DYT/PARK-SLC6A3 [3]. 

## 9. Therapeutic Perspectives

Inherited monoamine neurotransmitter disorders include a group of treatable diseases [115]. Almost complete control of movement disorders is achievable in some disorders, such as AD-DYT/PARK-GCH1, and a significant improvement in quality of life can be obtained with pharmacotherapy [115]. 

Table 3 summarizes the preferred therapeutic schemes for each of the abovementioned diseases. Available treatment strategies should be tailored to the specific neurotransmitter defect and include drugs inducing a replacement of depleted monoamine precursors, a decrease in monoamine catabolism, dopamine agonists, cofactors of monoamine metabolism, or a phenylalanine-restricted diet^115^. Robust evidence for these treatments’ efficacy is lacking; consequently, doses and the whole therapeutic strategies tend to rely more on expert recommendations [41,74,115]. 

Antisense oligonucleotide treatment resulted in the complete rescue of PTPS proteins in the fibroblasts of patients carrying two deep intronic variants activating pseudoexons on the *PTS* gene (164-672C > T and c.164-716A > T) [116]. More advanced progress was achieved with gene therapy in patients with DYT-DDC [117,118,119]. Thirty-nine patients received gene therapy through the stereotactic delivery of adenoviral vectors containing human *DDC* copies [117,118,119]. Thirty-three patients in Taiwan and Japan received injections in the putamen, while seven in the USA received injections in the substantia nigra and ventral-tegmental areas [117,118,119]. Both strategies resulted in dramatic and persistent motor improvements (especially in younger patients and lasting more than 5 years in the Taiwanese cohort), improved cognitive performances in patients with moderate phenotypes (in the Japanese cohort), and an increased dopamine production confirmed by CSF markers and PET scans/tractography in all patients [117,118,119]. 

Similar strategies based on lentiviral and adenoviral vectors provided the same promising results in patient-derived induced pluripotent stem cells and mouse models of DYT/PARK-SLC6A3 [120]. Enzyme activity restoration, the improvement of motor phenotypes, and the extension of lifespans and neuronal survival in the substantia nigra and striatum were observed in these experimental settings [120]. The injection of viral vectors in the midbrain of adult knock-out mice induced fewer toxic effects than intracerebroventricular administration [120]. 

## 10. Conclusions

Primary disorders of biogenic amine metabolism represent an interesting model of how clinical practice should be integrated with biochemical and molecular genetic techniques to improve patient outcomes in severe neurological disorders. Therapeutic progress, up to the enormous potentialities showed by gene therapy for DYT-DDC, will probably contribute to the achievement of this aim. 

## Figures and Tables

**Figure 1 genes-14-00263-f001:**
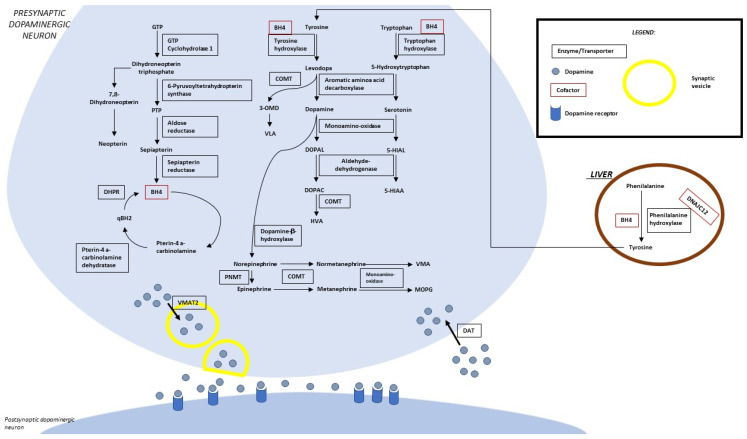
Biogenic amine and pterin metabolism. LEGEND: 5-HIAA, 5-hydroxyindoleacetic acid; 5-HIAL, 5-hydroxyindoleacetaldehyde; BH4, tetrahydrobiopterin; DHPR, dihydropterine reductase; DOPAC, 3,4-dihydroxyphenylacetic acid; DOPAL, 3,4-dihydroxyphenylacetaldehyde; DAT, dopamine transporter; GTP, guanosine-5′-triphosphate; HVA, homovanillic acid; MOPG, methoxylhydroxyphenylglycol; PLP, pyridoxal 5′-phosphate; PNMT, phenylethanolamine N-methyltransferase; PTP, 6-pyruvoyltetrahydropterin; qBH2, quinonoid dihydrobiopterin; VLA, vanillyllactic acid; VMA, vanillylmandelic acid; VMAT 2, vesicular monoamine transporter.

**Figure 2 genes-14-00263-f002:**
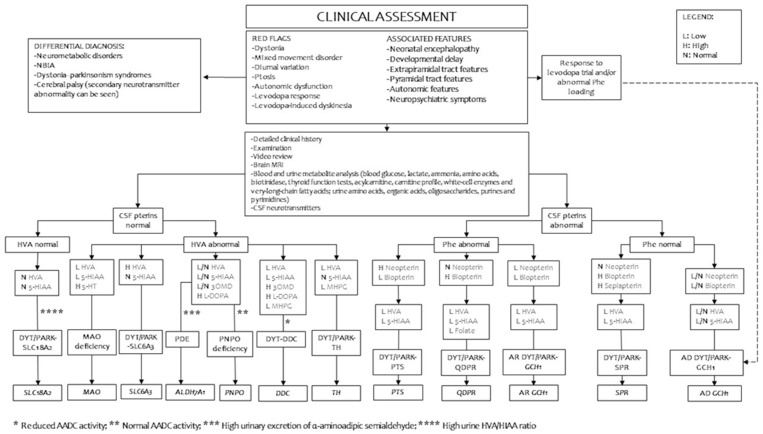
Suggested diagnostic work-up in patients with suspected defects of inherited monoamine neurotransmitter disorders. LEGEND: 3OMD, 3-O-methyldopa; 5-HIAA, 5-hydroxyindoleacetic acid; AADC, aromatic l-amino acid decarboxylase; AD, autosomal dominant; *ALDH7A1*, aldehyde dehydrogenase 7 family member A1; AR, autosomal recessive; CSF, cerebrospinal fluid; *DDC*, DOPA decarboxylase gene; *GCH1*, GTP cyclohydrolase 1 gene; *PTS*, 6-pyruvoyltetrahydropterin synthase gene; *QDPR*, quinoid dihydropteridine reductase gene; *SLC6A3*, solute carrier family 6 member A3, dopamine transporter gene; *SLC18A2*, solute carrier family 18 member A2, vesicular monoamine transporter 2 gene; *SPR*, sepiapterin reductase gene; *TH*, tyrosine hydroxylase gene; HVA, homovanillic acid; *MAO*, monoamine oxidase gene; MHPG, 3-methoxy-4-hydroxyphwnylglycol; NBIA, neurodegeneration with brain iron accumulation; *PNPO*, pyridoxamine 5’-phosphate oxidase gene.

**Table 1 genes-14-00263-t001:** Clinical and biochemical features of disorders of biogenic amine metabolism.

Disease	Gene	Clinical Features	Biochemical Markers		
			Plasma	Urine	CSF
AD-DYT/PARK-GCH1 (OMIM#128230)	*AD GCH1*	Parkinsonism, dystonia, motor delay, diurnal fluctuation, truncal hypotonia, hypertonia of extremities, tremors, and hypokinetic/rigid syndrome	Normal Phe Normal response to Phe loading	↓BIO, ↓NEO	↓NEO, ↓BIO, ↓HVA, ↓5-HIAA, Normal Sep and BH2
AR-DYT/PARK-GCH1 (OMIM#233910)	*AR GCH1*	Truncal hypotonia, parkinsonism, feeding/swallowing difficulties, dystonia, excessive sweating, temperature instability, intellectual disability, motor delay, choreoathetosis, drooling, oculogyric crises, ptosis, and seizures	↑Phe	↓BIO, ↓NEO	↓NEO, ↓BIO, ↓HVA, ↓5-HIAA, Normal Sep and BH2
DYT/PARK-PTS (OMIM#261640)	*PTS*	Dystonia, diurnal fluctuation, excessive sweating, temperature instability, hypo/hypertonia, parkinsonism, intellectual disability, motor delay, choreoathetosis, low birthweight, ptosis, and seizures	↑Phe ↑Prolactin	↓BIO, ↑NEO	↑NEO, ↓BIO, ↓HVA ↓5-HIAA, Normal Sep and BH2
DYT/PARK-SPR (OMIM#612716)	*SPR*	Motor delay, dystonia, truncal hypotonia, diurnal fluctuation, intellectual disability, parkinsonism, hypertonia, drooling, oculogyric crises, and hypokinetic/rigid syndrome	Normal Phe	↑Sep	Normal NEO, ↑BIO, ↓HVA, ↓5-HIAA ↑BH2, ↑Sep
DYT/PARK-QDPR (OMIM#261630)	*QDPR*	Dystonia, diurnal fluctuation, sweating, temperature instability, hypo/hypertonia, parkinsonism, intellectual disability, motor delay, choreoathetosis, low birthweight, ptosis, epileptic encephalopathy, and basal ganglia calcifications	↑Phe	↑BIO, ↓NEO	N NEO, ↑BIO, ↓HVA ↓5-HIAA, ↓Folate, ↑BH2 Normal Sep
PCBD deficiency (OMIM#264070)	*PCBD1*	No severe neurological symptoms and transient and benign hyperphenylalaninemia	↑Phe	↑Primapterin	/
DYT/PARK-TH (OMIM#605407)	*TH*	Hypokinetic/rigid syndrome, dystonia/parkinsonism, oculogyric crises, ptosis, autonomic dysfunctions, lethargy, irritability, sleep disturbances, pre-term birth, foetal distress, perinatal asphyxia, intellectual disability, growth retardation, microcephaly, motor delay, spasticity, and myoclonus	Normal Phe	/	↓HVA, Normal NEO, BIO, 5HIAA, Sep, BH2
DYT-DDC (OMIM#608643)	*DDC*	Truncal hypotonia, developmental delay, intellectual disability, oculogyric crises, dystonia, dysarthria, ptosis, limb hypertonia, choreoathetosis, sleep disturbances, excessive sweating, temperature instability, orthostatic hypotension, diarrhea, nasal congestion, and hypoglycemia	Normal Phe ↑ Prolactin	↑ Dopamine, ↓ VMA ↑ Vanillactic acid	↓HVA, ↓5-HIAA, ↑3OMD ↓MHPG, Normal NEO, BIO, Sep, BH2
MAOA/MAOB deficiency (OMIM#300615 MAOA)	*MAOA/* *MAOB*	Behavioral disturbances, mild intellectual disability, hand stereotypes, flushing, and diarrhea	/	↑normetanephrin ↑3-methoxytyramine, ↑tyramine ↓VMA, ↓HVA, ↓MHPG, ↓5-HIAA	/
DBH deficiency (OMIM#223360)	*DBH*	Severe orthostatic hypotension, eyelid ptosis, sporadic dysmorphic features, rare reproductive dysfunctions, and normal cognitive development	↓Norepinefrine ↑ Dopamine	/	/
DYT/PARK-SLC6A3 (OMIM#613135)	*SLC6A3*	Severe developmental delay or no acquisition of developmental milestones, anarthria, dystonia, parkinsonism, dyskinesia, oculogyric crises, swallowing difficulties, failure to thrive, and respiratory complications	Normal Phe ↑Prolactin ↓Norepinefrine	↓Norepinefrine ↑3MT	↑ HVA, Normal NEO, BIO,5-HIAA, Sep, BH2
DYT/PARK-SLC18A2 (OMIM#618049)	*SLC18A2*	Severe developmental delay or no acquisition of developmental milestones, dysarthria, dystonia, parkinsonism, facial dyskinesia, oculogyric crises, vertical gaze palsy, and ptosis	/	↑ HVA, ↑5-HIAA, ↓Dopamine, ↓Norepinefrine	↑HVA/5-HIAA
DNAJC12 deficiency (OMIM#617384)	*DNAJC12*	Juvenile parkinsonism, dystonia, autism, intellectual disability, attention deficit hyperactivity disorder, psychiatric symptoms, and no symptoms in a quote of patients	↑Phe	/	↑BH4 ↓HVA, ↓5-HIAA

LEGEND: 3MT, 3-methoxytyramine; 5 HIAA, 5-hydroxyindolacetic acid; *GCH1*, GTP Cyclohydrolase 1 gene; BIO, biopterin; BH2, dihydrobiopterin; *DBH*, dopamine β hydroxylase gene; *DNAJC12*, DNAJ/HSP40 homolog, subfamily c, member 12; *DDC*, DOPA decarboxylase gene; *DYT/PARK-PTS*, 6-pyruvoyltetrahydropterin synthase gene; *QDPR*, quinoid dihydropteridine reductase gene; *SLC6A3*, solute carrier family 6 member A3, dopamine transporter gene; *SLC18A2*, solute carrier family 18 member A2, vesicular monoamine transporter 2 gene; *SPR*, sepiapterin reductase gene; *TH*, tyrosine hydroxylase gene; HVA, homovanillic acid; *MAO A/B*, monoamine oxydase A/B gene; MHPG, 3-methoxy-4-hydroxyophenylglycol; NEO, neopterin; *PCBD1*, pterin-4a-carbinolamine dehydratase gene; Phe, phenylalanine; Sep, sepiapterin; VMA, vanillylmandelic acid. ↓ = decreased;↑ = increased.

**Table 2 genes-14-00263-t002:** Interpretation of variants with the most significant clinical implications, as discussed in the text, according to the ClinVar (https://www.ncbi.nlm.nih.gov/clinvar/; last access 7 October 2022) and BioPKU (http://www.biopku.org/; last access 7 October 2022) databases.

VARIANT	BioPKU	ClinVar
** *NM_000161.3* ** ** *(GCH1)* **		
c.218C > A; p.(Ala73Asp)	VUS	N/A
c.309G > C; p.(Gln103His)	VUS	N/A
c.617T > C; p.(Val206Ala)	VUS	VUS
c.633G > A; p.(Met211Ile)	Pathogenic	Pathogenic (no assertion criteria)
c.703C > T; p.(Arg235Trp)	VUS	N/A
c.166G > A; p.(Glu56Lys)	VUS	N/A
c.638T > C; p.(Met213Thr)	VUS	N/A
c.350T > G; p.(Leu117Arg)	VUS	N/A
c.265C > T; p.(Gln89*)	VUS	N/A
c.262C > T; p.(Arg88Trp)	VUS	Pathogenic with no assertion criteria
c.630C > G; p.(His210Gln)	VUS	N/A
c.557C > A; p.(Thr186Lys)	VUS	N/A
** *NM_000317.3(PTS)* **		
c.84-291A > G	N/A	Conflicting interpretations as VUS, likely pathogenic, or pathogenic
c.58T > C; p.(Phe20Leu)	N/A	N/A
c.46C > T; p.(Arg16Cys)	N/A	VUS
c.338A > G; p.(Tyr113Cys)	N/A	Conflicting interpretations as pathogenic, likely pathogenic, or VUS
c.370G > T/; p.(Val124Leu)	N/A	Conflicting interpretations as pathogenic or likely pathogenic
c.139A > G; p.(Asn47Asp)	N/A	Pathogenic with no assertion criteria
c.347A > G; p.(Asp116Gly)	N/A	Likely pathogenic
c.412A > C; p.(Asn138His)	N/A	N/A
c.120T > G; p.(Phe40Leu)	N/A	N/A
c.216T > A; p.(Asn72Leu)	VUS	VUS
c.430G > C; p.(Gly144Arg)	N/A	N/A
c.385A > G; p.(Lys129Glu)	N/A	N/A
c.46C > T; p.(Arg16Cys)	N/A	VUS
** *NM_000320.3(QDPR)* **		
c.451G > A; p.(Gly151Ser)	VUS	N/A
c.635T > G; p.(Phe212Cys)	VUS	N/A
c.199-1G > T	N/A	VUS
** *NM_003124.5(SR)* **		
c.512G > A; p.(Cys171Tyr)	VUS	Likely pathogenetic
c.68G > A; p.(Gly23Asp)	N/A	N/A
c.207C > G; p.(Asp69Glu)	VUS	VUS
c.-13G > A	VUS	Pathogenic with no assertion criteria
** *NM_000360.4(TH)* **		
c.605G > A; p.(Arg202His)	N/A	Pathogenic with no assertion criteria
c.614T > C; p.(Leu205Pro)	N/A	Pathogenic/likely pathogenic
c.-71C T	N/A	Conflicting interpretations as pathogenic versus likely pathogenic
c.646G > A; p.(Gly216Ser)	N/A	Conflicting interpretations as pathogenic versus likely pathogenic
c.983G > T; p.(Cys328Phe)	N/A	Likely pathogenic
** *NM_001082971.2(DDC)* **		
c.478C > T; p.(Arg160Trp)	VUS	Conflicting interpretations as likely pathogenic or VOUS
** *NM_000240.4(MAO-A)* **		
c.1336G > A; p.(Glu446Lys)	N/A	N/A
c.886C > T; p.(Gln296Ter)	N/A	Pathogenic with no assertion criteria
** *NM_001044.5(SLC6A3)* **		
c.561C > T; p.(Arg521Trp)	N/A	VUS
c.934A > T; p.(Ile312Phe)	N/A	N/A
c.1261G > A; p.(Asp421Asn)	N/A	N/A
c.1639dupC; p.(His547Profs*56)	N/A	Likely pathogenic
** *NM_003054.4(SLC18A2)* **		
c.710C > A; p.(Pro237His)	N/A	VUS
c.127A > T; p.(Ile43Phe)	N/A	N/A
c.1160C > T; p.Pro387Leu	N/A	Pathogenic with no assertion criteria
** *NM_021800.3(DNAJC12)* **		
c.306C > G; p.(His102Gln)	N/A	N/A
c.182delA/p.Lys61Argfs*6	N/A	N/A
c.85delC/p.Gln29Lysfs*38	N/A	N/A
c.298-968_503-2603del	N/A	Pathogenic with no assertion criteria
c.596G > T; p.(Ter199Leu)	N/A	Pathogenic with no assertion criteria
c.214C > T/p.(Arg72Ter)	N/A	Pathogenic with no assertion criteria

VUS= variants of unknown significance; N/A= not available.

**Table 3 genes-14-00263-t003:** Available pharmacological treatments for inherited biogenic amine metabolism disorders.

Disorder	Disease	Therapy
**Co-chaperone defects**	DNAJC12 deficiency	BH4: 1–10 mg/kg/d; L-Dopa combined with carbidopa (4:1 ratio): 1–10 mg/kg/d; 5-HTP: 1–10 mg/kg/d.
**Biopterin synthesis/recycling defects**	DYT/PARK-SPR	L-Dopa combined with carbidopa (4:1 ratio): 1–10 mg/kg/d; 5-HTP: 1–8 mg/kg/d; Selegiline: 0.03 to 0.2 mg/kg/d.
	AD-DYT/PARK-GCH1	L-Dopa combined with carbidopa (4:1 ratio): 1–10 mg/kg/d.
	AR-DYT/PARK-GCH1	L-Dopa combined with carbidopa (4:1 ratio): 1–10 mg/kg/d; 5-HTP: 1–6 mg/kg/d; BH4: 1–10 mg/kg/d
	DYT/PARK-PTS	L-Dopa combined with carbidopa (4:1 ratio): 0.5–1 mg/kg/d; 5-HTP: 1–8 mg/kg/d; BH4: 1–10 mg/kg/d
	DYT/PARK-QDPR	L-Dopa: 0.5–10 mg/kg/d; 5-HTP: 3–11 mg/kg/d; Diet to control Phenylalanine levels; Folinic acid: 10–20 mg/d
	PCBD deficiency	BH4: titrated according to Phe levels
**Primary neurotransmitter synthesis defects**	DYT/PARK-TH	L-Dopa combined with carbidopa (4:1 ratio): 0.5–10 mg/kg/d; Selegiline: 0.1–0.4 mg/kg/d (max. dose 10 mg/d)
	DYT-DDC	**Dopamine agonists:** Pramipexole BASE: 5–10 µg/kg/d (max. 75 µg/kg/d); Ropinirole: 0.25 mg/d (max. 24 mg/d); Transdermal rotigotine: 2–8 mg/d; Bromocriptine: 0.1–0.5 mg/kg/d. **MAO inhibitors:** Selegiline: 0.1–0.3 mg/kg/d; Tranylcypromine: 0.1–30 mg/kg/d. **Co-factors:** Pyridoxine: 100–200 mg/d; Pyridoxal 5′-phosphate: 100–200 mg/d
**Monoamine transportopathies**	DYT/PARK-SLC6A3	Pramipexole BASE: 5–40 µg/kg/d; Ropinirole: 0.5–4 mg/d; Transdermal rotigotine: 6 mg/kg/d
	DYT/PARK-SLC18A2	Pramipexole BASE: 5–40 µg/kg/d
**Monoamine catabolism disorders**	MAOA/MAOB deficiency	SSRI have shown beneficial effect in mice, no data for humans available
	DBH deficiency	Droxidopa (adults): 100 mg 3 times a day (max. 600 mg 3 times a day); No data for pediatric use available

LEGEND: 5-HTP, 5-hydroxytryptophan; AD-DYT/PARK-GCH1, autosomal dominant GTP cyclohydrolase 1; AR-DYT/PARK-GCH1, autosomal recessive GTP cyclohydrolase 1; BH4, tetrahydrobiopterin; *DBH*, dopamine β hydroxylase; *DNAJC12*, DNAJ/HSP40 homolog, subfamily c, member 12; DYT-DDC, DOPA decarboxylase deficiency; DYT/PARK-PTS, 6-pyruvoyltetrahydropterin synthase; DYT/PARK-QDPR, quinoid dihydropteridine reductase; DYT/PARK-SLC6A3, solute carrier family 6 member A3, dopamine transporter; DYT/PARK-SLC18A2, solute carrier family 18 member A2, vesicular monoamine transporter 2; DYT/PARK-SPR, sepiapterin reductase; DYT/PARK-TH, tyrosine hydroxylase; MAO A/B, monoamine oxydase A/B; *PCBD1*, pterin-4a-carbinolamine dehydratase; Phe, phenylalanine; SSRI, selective serotonin reuptake inhibitor.

## Data Availability

Appendix A contains the collected data.

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
