# Peer review of "Phenotypes and Genotypes of Inherited Disorders of Biogenic Amine Neurotransmitter Metabolism"

_genes, 2023, doi:10.3390/genes14020263_

Round 1

Reviewer 1 Report

This is overall an interesting and helpful review. Overall, I would suggest that the authors go through the manuscript extensively to check the editing and formatting and make it easier to read.

- The authors have used the MDS-Gene nomenclature system and should perhaps include the relevant citation for the reader (see pubmed for numerous examples). In this case, only the gene name should be in italics – e.g. for ‘DYT/PARK-GCH1’ – only ‘GCH1’ should be in italics. The author could also refer to this helpful review and respective online database - PMID: 34908184 – when discussing dopa-responsive dystonia. Also, please note that ‘DYT/PARK-GCH1’ is the MDSGene recommended name for the disorder, and is not the name of the gene, the name of the gene is ‘GCH1’. This should be corrected for all disorders throughout the text.

- The terms ‘likely pathogenic’ or ‘pathogenic’ can refer to the ACMG criteria (both ‘likely pathogenic’ and ‘pathogenic’ may be considered diagnostic) – perhaps instead using ‘disease-causing’ or ‘clinically-relevant’ variants when not referring specifically to the ACMG criteria.

- The individual variants discussed could be included in a table format which would be easier for the reader to refer to and should then be removed from the text. This would free up the text to discuss genotype-phenotype correlation, clinical pearls and diagnostic clues, treatment implications, etc. which should be discussed in greater detail.

- The table and figures were not readily accessible for review.

- Consider using examples of MRI and clinical pictures etc. to illustrate important findings to the reader, may be especially important when the authors discuss their own personal cases.

- I would suggest using ‘heterozygous’ rather than ‘heterozygosis’.

- OMIM numbers do not need to be in the subtitles, they are already in the text.

- Further English language editing throughout the manuscript would be helpful, examples below listed below each section.

Clinical presentation

- What do the authors mean by ‘scholar age’.

- What do the authors mean by ‘neurological derangement’ – I would suggest another term.

- ‘contest of an infantile encephalopathy’ should be ‘context of an infantile encephalopathy’

- The dot points mentioned here are confusing from the perspective of an adult neurologist – e.g. first there is mention of ‘hypotonia associated with dystonia pattern’, then the opposite ‘rigidity associated with dystonic postures’ (do they mean lead-pipe?), it is unclear what ‘anti-gravity reactions’ are.

For section 4.1.1.

- ‘Dystonia and tone abnormalities’ are not symptoms.

For section 4.1.2.

- ‘read pipe’ should be ‘lead pipe’

For section 4.2.1.

- It may be confusing to the reader when ‘a compound hetozygosis state’ (?heterozygous) is mentioned in a section on autosomal dominant DYT/PARK-GCH1.

- The dot points could be presented in table form.

For section 4.2.3.

-          It should be peak dose dyskinesa rather than ‘pick dose’.

For section 6.2.

- ‘Uncomplete’ should be ‘incomplete’.

For section 7.1.

- What do the authors mean by ‘homonym gene’?

DIAGNOSTIC WORK-UP

- Please check c.105delC/ (c.710 T>C – there should be no space between c.710 and ‘T>C’ (please check the manuscript for this), also remove the bracket.

Author Response

REVIEWER 1

This is overall an interesting and helpful review. Overall, I would suggest that the authors go through the manuscript extensively to check the editing and formatting and make it easier to read.

We would acknowledge the reviewer for his/her positive judgement and for his/her valuable comments and suggestions.

The authors have used the MDS-Gene nomenclature system and should perhaps include the relevant citation for the reader (see pubmed for numerous examples). In this case, only the gene name should be in italics – e.g. for ‘DYT/PARK-GCH1’ – only ‘GCH1’ should be in italics. The author could also refer to this helpful review and respective online database - PMID: 34908184 – when discussing dopa-responsive dystonia. Also, please note that ‘DYT/PARK-GCH1’ is the MDSGene recommended name for the disorder, and is not the name of the gene, the name of the gene is ‘GCH1’. This should be corrected for all disorders throughout the text.

The italics character was maintained for gene names only and the suggested reference was inserted. The name of the genes was corrected throughout the text (without the prefix DYT/PARK- that was maintained for the name of the disorders.

The terms ‘likely pathogenic’ or ‘pathogenic’ can refer to the ACMG criteria (both ‘likely pathogenic’ and ‘pathogenic’ may be considered diagnostic) – perhaps instead using ‘disease-causing’ or ‘clinically-relevant’ variants when not referring specifically to the ACMG criteria.

The abovementioned terms for the identification of the variants referred directly or indirectly to ACMG criteria in most of the cases throughout the text. According to the reviewer’s suggestion, we replaced them with the term “disease-causing” in the few parts in which the link with ACMG criteria could be considered as weaker (page 7, line 18; page 11 line 22; page 12 line 9).

- The individual variants discussed could be included in a table format which would be easier for the reader to refer to and should then be removed from the text. This would free up the text to discuss genotype-phenotype correlation, clinical pearls and diagnostic clues, treatment implications, etc. which should be discussed in greater detail.

We aimed to report the main genotype-phenotype correlations about biogenic amine neurotransmitter metabolism because we considered these aspects as relevant topics for a journal like “Genes”. The discussion about individual variants was maintained in the text when it had clinical implications because we think that these aspects have a remarkable interest for the readers of the journal (including several genetists). We removed from the main body and inserted in a separate Table the interpretation of pathogenicity according to ClinVar and BiopKu databases to improve the readability of the text.

 A more detailed discussion about clinical, diagnostic, and therapeutic implications was beyond the scope of this review because of they were already analysed in relatively recent valuable review articles.

- The table and figures were not readily accessible for review.

We apologize for this inconvenience. We had correctly uploaded the related files on the website. We do not know the reason of their lacking accessibility. We would suggest asking the Editorial staff of the journal to provide the related files for review in case of persisting problems of visualization.

-Consider using examples of MRI and clinical pictures etc. to illustrate important findings to the reader, may be especially important when the authors discuss their own personal cases.

The suggestion is correct but two important questions should be considered:

-the MRIs of our personal cases were generally uninformative (as in most of the published cases in literature apart from the data that had already been discussed in another part of the text)

-the complete clinical pictures of our cited cases were already published elsewhere, and the related references had been provided (ref. 54 and 69). We added details about carried variants and affected relatives of the cited patient with autosomal dominant-DYT/PARK-GCH1 (reference 54)

I would suggest using ‘heterozygous’ rather than ‘heterozygosis’.

The text was modified accordingly

OMIM numbers do not need to be in the subtitles, they are already in the text.

OMIM numbers were removed from the subtitles

Further English language editing throughout the manuscript would be helpful, examples below listed below each section.

The text was deeply revised for English syntax and style by a native speaker

- What do the authors mean by ‘scholar age’.

The incorrect term was replaced with “school age”

- What do the authors mean by ‘neurological derangement’ – I would suggest another term.

The term was replaced with “neurological impairment”.

- ‘contest of an infantile encephalopathy’ should be ‘context of an infantile encephalopathy’

The typing error was corrected accordingly

-- The dot points mentioned here are confusing from the perspective of an adult neurologist – e.g. first there is mention of ‘hypotonia associated with dystonia pattern’, then the opposite ‘rigidity associated with dystonic postures’ (do they mean lead-pipe?), it is unclear what ‘anti-gravity reactions’ are.

We eliminated the dot points and we re-wrote this part as parts of a unique text to present the main pediatric peculiarities of movement disorders  in the disorders of biogenic amine metabolism (page 6.; lines 9-22).

.

Dystonia and tone abnormalities’ are not symptoms.

We corrected accordingly (we indicated them as signs)

- ‘read pipe’ should be ‘lead pipe’

We corrected accordingly

- It may be confusing to the reader when ‘a compound hetozygosis state’ (?heterozygous) is mentioned in a section on autosomal dominant DYT/PARK-GCH1.

We removed the term “compound” (the cited part was focused on dominant negative effect).

- The dot points could be presented in table form.

We preferred to incorporate them as parts of the text (without listing them as dot points) in almost all the paper. Dot points were maintained in the section “Autosomal Dominant DYT/PARK-GCH1” because they well summarized the reported early clinical presentations.

-          It should be peak dose dyskinesa rather than ‘pick dose’.

We changed accordingly

‘Uncomplete’ should be ‘incomplete’.

We changed accordingly

- What do the authors mean by ‘homonym gene’?

We inserted the name of DNAJC12 gene

Please check c.105delC/ (c.710 T>C – there should be no space between c.710 and ‘T>C’ (please check the manuscript for this), also remove the bracket.

We changed accordingly

Reviewer 2 Report

Genes Review:

Line 21: Please Change the sentence from “The earlier the onset of the disease is, the more severe and generalized is the derangement of motor functions.” into: “The earlier the disease manifests itself, the more severe and widespread the impairment of motor functions.”

Line 27 and 28: Please Change into the following for better readability:

The rarity of these diseases, combined with limited knowledge of their clinical, biochemical, and molecular genetic features, frequently leads to misdiagnosis or significant diagnostic delays.

Line 61: introduce space: “above mentioned”

This is a very interesting review with many genetically impaired biogenic amine pathways. To understand the complex process, it will be simple to present an interconnected metabolic pathways flow chart involved in these gene disorders.

Author Response

Line 21: Please Change the sentence from “The earlier the onset of the disease is, the more severe and generalized is the derangement of motor functions.” into: “The earlier the disease manifests itself, the more severe and widespread the impairment of motor functions.”

We changed it accordingly

Line 27 and 28: Please Change into the following for better readability:

The rarity of these diseases, combined with limited knowledge of their clinical, biochemical, and molecular genetic features, frequently leads to misdiagnosis or significant diagnostic delays.

We changed it accordingly

This is a very interesting review with many genetically impaired biogenic amine pathways. To understand the complex process, it will be simple to present an interconnected metabolic pathways flow chart involved in these gene disorders.

We would acknowledge the reviewer for his/her positive judgement and for his/her valuable comments. We had already submitted on the website a figure representing the metabolic cascade involved in these disorders (Fig.1). Another reviewer wrote that Figures and Tables were not accessible for review, and we do not know the reasons. If you had similar problems, we would suggest asking the Editorial staff of the journal to provide the related files for the following review steps.